# Blockchain-Based Securing of Data Exchange in a Power Transmission System Considering Congestion Management and Social Welfare

**Moslem Dehghani** [1], **Mohammad Ghiasi** [1], **Taher Niknam** [1,*], **Abdollah Kavousi-Fard** [1], **Mokhtar Shasadeghi** [1], **Noradin Ghadimi** [2,3,*] and **Farhad Taghizadeh-Hesary** [4]

1   Department of Electrical and Electronics Engineering, Shiraz University of Technology, Shiraz 71555/313, Iran; dehghani.kau@gmail.com (M.D.); m.ghiasi1@gmail.com (M.G.); abdollah.kavousifard@gmail.com (A.K.-F.); shasadeghi@sutech.ac.ir (M.S.)
2   Young Researchers and Elite Club, Ardabil Branch, Islamic Azad University, Ardabil 19585/466, Iran
3   Department of Industrial Engineering, Ankara Yıldırım Beyazıt University (AYBU), 06760 Ankara, Turkey
4   Social Science Research Institute, Tokai University, Kanagawa, Hiratsuka 259-1292, Japan; farhad@tsc.u-tokai.ac.jp
*   Correspondence: niknam@sutech.ac.ir (T.N.); nghadimi@ybu.edu.tr (N.G.)

**Abstract:** Using blockchain technology as one of the new methods to enhance the cyber and physical security of power systems has grown in importance over the past few years. Blockchain can also be used to improve social welfare and provide sustainable energy for consumers. In this article, the effect of distributed generation (DG) resources on the transmission power lines and consequently fixing its conjunction and reaching the optimal goals and policies of this issue to exploit these resources is investigated. In order to evaluate the system security level, a false data injection attack (FDIA) is launched on the information exchanged between independent system operation (ISO) and under-operating agents. The results are analyzed based on the cyber-attack, wherein the loss of network stability as well as economic losses to the operator would be the outcomes. It is demonstrated that cyber-attacks can cause the operation of distributed production resources to not be carried out correctly and the network conjunction will fall to a large extent; with the elimination of social welfare, the main goals and policies of an independent system operator as an upstream entity are not fulfilled. Besides, the contracts between independent system operators with distributed production resources are not properly closed. In order to stop malicious attacks, a secured policy architecture based on blockchain is developed to keep the security of the data exchanged between ISO and under-operating agents. The obtained results of the simulation confirm the effectiveness of using blockchain to enhance the social welfare for power system users. Besides, it is demonstrated that ISO can modify its polices and use the potential and benefits of distributed generation units to increase social welfare and reduce line density by concluding contracts in accordance with the production values given.

**Keywords:** FDIA; blockchain; data exchanging; under-operating agents; ISO; electricity market

## 1. Introduction

As a result of the advancement of technology in power systems, the importance of using new methods to protect smart grids (SGs) becomes more apparent, and one of these methods is blockchain. In October 2008, in the reference [1], blockchain technology distribution was seen under the alias "Satoshi Nakamoto", with the aim of supporting the first Bitcoin cryptocurrency, and it resulted in the start of the Bitcoin network in January 2009. After that, Bitcoin has arrived inchmeal at the financial industry systems, and is recognized as the most influential and important cryptocurrency. Besides, blockchain technology after Bitcoin has become a game-variable innovation around the world, and lots of industries exist which will be interrupted via blockchain, including the life sciences, legal industry, health care, financial services, cyber security, supply series management,

cloud storage, charity, electing, government, social interests, energy management, private transport and ride sharing, retail, and actual estate between others [2–4]. Blockchain usages in carbon credits, distributed energy resources, and renewable and power system data security versus cyber-attacks are really encouraging [5,6].

So, the available electricity market must be modified to benefit from this novel technology. In this study, there are two purposes, including examining how a great industrial user is able to manage best under such a novel blockchain on the basis of an electricity market as well as how to be safe from cyber-attacks like false data injection attack (FDIA).

### 1.1. Background

Networking, informatization, and intelligentization have been gradually realized by power systems [7] with the help of the usage of advanced information technology like calculation, communication, control, and perception, as well as using the expansion of the cyber-physical power system (CPPS). Meanwhile, real-time analytical promotion, efficient allocation of power systems and scientific decisions, interface terminals, and the open communication networks also carry potential security risks [8,9]. In comparison with the relatively strong initial power system, investigations on the security protection of electrical power information networks are in the initial stages, with a great deal of security vulnerabilities unknown [10]. Due to the remarkable interest relevance and high transmissibility of an electrical power grid, after the attack, it will have a rigid effect on industrial production, power security, and the livelihood of people, which has attracted considerable regard [11]. As a new attack method for major industrial facilities, safe and stable power system operation, whose defense and attack structure needs further research and investigation [12], have been threatened by cyber-attack.

Pursuant to the purposes of the attack, cyber-attacks versus power systems are able to be categorized as integrity, confidentiality of information, and destruction of availability. The availability destruction is obtained in attacks modifying network topology, black hole attacks, and unavailable information due to communication interruption, whose common layouts are denial of service (DOS) attacks. The integrity destruction is obtained in replay attack, man-in-the-middle attack, and wrong information due to false data injection (FDI), whose common layouts are FDI attacks. The confidentiality destruction is obtained in internal employee attack, utilization of malware, and illegal usage and data leakage, whose common layout is brute force password cracking [13,14]. As a common way to destroy the integrity of information, the FDIA is able to interrupt the outcomes of state estimation analysis, therefore misleading the control center's decision. The FDIA was first introduced by Liu in 2009, and the FDIA's point is to conduct coordinated attacks on sensors, falsify specific measurements, and manipulate particular state estimation information [15,16].

Over the past decade, the concept of the micro-grid (MG) is gaining more popularity resulting from its economic and technical benefits like higher reliability and resilience, closeness to the users, higher power quality, lower power losses, less operation cost, and self-healing capability [17,18]. Nevertheless, along with these benefits, several significant challenges exist in the management and operation of the micro-grids (MGs), which have attracted the attention of plenty of researchers [19]. In the ordinary power network, the distributed system operator (DSO) is responsible for applying the optimal energy management throughout the grid.

Nevertheless, in recent power networks, DSOs and MGs might have various policies and owners. As the network is able to be formed of some intercoupled MGs, the total network operation can be significantly affected by any change in any of the subsystems. So, a coordinator of a strong system level is required for the total network energy management.

Developments in management and economic laws in the electricity industry have long been taking place in this large industry under the congestion of the power system structure. The problem of congestion of the transmission network is considered as one of the most important issues in the discussion of the full implementation of the restructuring of power systems. Congestion means using a transmission network outside the operating range.

Restrictions such as transmission line capacity and transformers, maximum and minimum voltage values, and maximum voltage angles of the busbars (which are determined by various studies on the network) are among the limitations of operation. The consequences of congestion in power systems include sudden price jumps in some areas, increased market power, increased electricity prices, reduced efficiency of the electricity network, reduced competition, etc. [20,21]. When the power grid is congested, the capacity of some transmission lines no longer meets the needs of all customers. In this case, the independent system operator, as the main institution for maintaining the security of the power system, acts in different ways to manage the network's congestion. Independent system operation (ISO) is an institution that is responsible for coordinating and maintaining the security and reliability of power systems; in this regard, density management is the main task of this institution, which always tries to encourage investors and private owners of electric generators to participate in density management [22].

Congestion may be demonstrated during operation or in network operation planning. When using the power system, factors such as the sudden departure of one or more transmission lines or transformers in the network, the unexpected cessation of production in one or more generators in the system, and unforeseen changes in energy consumption, as well as uncoordinated transactions in electricity sales, can lead to network congestion. In planning the operation of a power system, the provision of an inappropriate program for the generation and consumption of electricity, the implementation of which violates the restrictions on the operation of the transmission network, is the main cause of network congestion [23].

Lately, blockchain has been promoted as a reliable and effective technology for online financial operations via communication only between peer-to-peer transaction networks and without the intervention of third parties [24]. The data are able to be reserved in the distributed databases, which are generally small, instead of reserving all data in a central data center by using blockchain. This might result in increasing the total cloud system security, since more of the losses from attacks on such databases are simply able to be locally prevented. So, blockchain is able to be successfully used in different areas, like the Internet of Things (IoT) and the financial sector.

Peer-to-peer (P2P) energy trading research based on blockchain has been studied. In [25], a blockchain with seven components based on MG energy market was offered and a smart contract was used to make a high-performance information system on the basis of the blockchain strategy. In [26], blockchain has been used to constitute a machine to machine electricity market in a chemical industry context, and a private blockchain on the basis of software system multichain was applied to validate energy transaction.

With the aim of facilitating a P2P market, a two-layer energy market system on the basis of multi-agent and blockchain technology has been suggested in [27]. For electric vehicles, also, a blockchain-based consortium for local aggregators has been proposed in [28] with the aim of auditing and validating electricity market plug-in hybrid electric vehicles (PHEVs). A novel blockchain-based energy system has been suggested in [29] with the aim of enabling electric vehicles to share transactions and publicly audit without the support of any reliable interposition. Plus, the consortium blockchain method was expanded to generic energy blockchain transactions with the aim of transaction security and credit-based payment. In [30], a decentralized energy dealing system using blockchain on the basis of tokens, streams of unknown encrypted messaging, and multi-signatures is proposed with the aim of solving issues of the privacy and security of information of demand and dealing.

In [31], forward and real-time markets have been proposed for bilateral agreements in P2P energy dealing. In [32], the energy broker's role has been proposed via amplification learning for indirect energy trading between customers. For electric vehicles, [33], a special P2P trading system was studied with the aim of reducing the effect of the charging procedure during the working hours of power systems. Additionally, in [34], in P2P energy trading, different game– and auction–theoretic methods were studied, and in [35],

especially, a Nash bargaining view was applied with the aim of developing a frame of bilateral trans-active energy dealing for numerous contributors. For P2P trading, in [36], restrictions of physical low-voltage networks have been investigated by applying sensitivity analysis. Furthermore, in [37], power damages were assigned for MG peer-to-peer blockchain market contributors. At the power distribution system level, [38] suggested a day-ahead forecasting energy market strategy to help distribution system operators in order to optimize distributed energy resource applications; however, in [39], a novel P2P energy market on the basis of the content of multi-class energy management with the aim of coordinating dealing amongst prosumers with heterogeneous preferences was introduced.

Blockchain can be considered as the main technology of Bitcoin and some other types of cryptocurrencies, making this one of the world's most breathtaking technologies in 2010. The problem of transmission network conjunction has always been one of the serious obstacles against the full implementation of the restructuring of power systems, the correct and free communication of the producers and consumers, and the main challenge of independent system operation. Therefore, the policy of using distributed generation resources to manage the conjunction of transmission lines has been of particular importance. The policy of applying the transparent blockchain technology will aid in reducing risks and provide superior security to the grid, hence financial fraud is eliminated and the entire operational charge is attenuated. In order to illustrate the matters occurring in the usual blockchain layouts, mainly because of the storage and high level of elaborations of hash address computations, a blockchain technology is proposed in this article. Plus, a new data restoration technique is expanded with the aim of providing an approach to restore the appropriate and accurate data.

Nowadays, with the use of communication networks to exchange information between ISO and under-operating agents, sabotage cyber-attacks, including the false data injection attack (FDIA), are on the rise in order to destroy network stability and inflict financial losses. Therefore, providing a solution and policy to secure the information exchanged between systems is of great importance, so the use of blockchain technology as a solution to secure data is essential.

In this paper, the system under study is first examined under various conditions, including normal mode and false data injection into exchanged data such as loads, prices, and productions, and it is shown that this cyber-attack disrupts network congestion, reduces welfare, increases costs, and upsets the balance of production and demand.

### 1.2. Paper Structure

The remainder of the article is presented as follows: Section 2 defines the principles of blockchain and false data injection attacks. Problem theory, along with the formulation and the network under study and the system parameters, are presented in Section 3. The simulation results are analyzed under different normal scenarios and false data attacks and blockchain techniques to secure the messages and data in Section 4. Finally, the final conclusion is given in Section 5.

## 2. Basic Concept

### 2.1. The Data-Sharing Structure According to Blockchain Technology

#### 2.1.1. Framework Overview

In order to present services based on a fine-grained and data-sharing structure, the transactions saved in the blockchain database based on the privacy level have been classified. The level of privacy contains community data, encrypted data, and public data. Public data, here, refer to a datum which is able to be observed through entire nodes. In addition, community public data give information which is able to be recognized through entire nodes pertaining to the similar community, and encrypted information basically refers to the private data and those which users are willing to purchase/sell.

In general, when users share professional information, it is suggested that they adjust the amount of information privacy on public data in the community so that this informa-

tion is shared by more users who actually access it, and that need it to be visible. As a result, the main purpose can be to apportion the community reasonably through gaining a community diagnosing technique so that the public information of the community is able to be divided between more users who actually require it. Figure 1 depicts the data-sharing structure according to the blockchain method. We have three different layers in our presented data-sharing strategy, such as Data Layer, Blockchain Layer, and Detection Layer. Information is gathered by the Data Layer and sent to the Detection Layer. The Detection Layer implements a community detection technique that divides customers into various communities and limits the domain of data sharing. The Blockchain Layer also has the responsibility of keeping the result of community detection and transaction records secure.

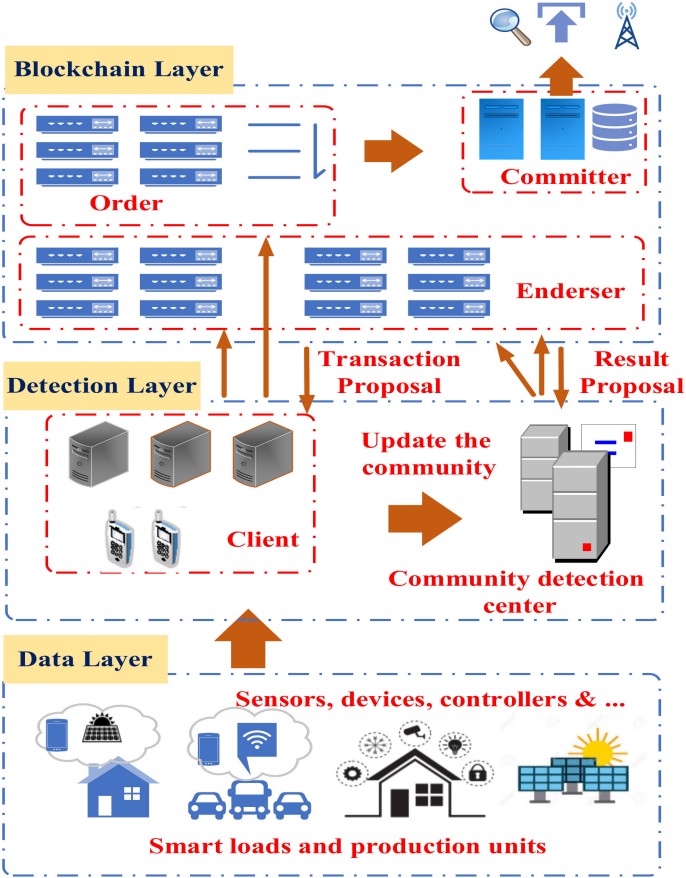

**Figure 1.** The data-sharing structure according to the blockchain technology.

### 2.1.2. Data Layer

The Data Layer contains data perceived through large-scale sensors [40] such as product type and product quantity, device performance status, and other data of several parameters.

Perception data are obtained by sensors and uploaded to the client server for comprehensive analysis and sharing. Perception information, after being obtained from the sensors, is sent and shared on the client's server for analysis.

### 2.1.3. Detection Layer

The Detection Layer includes the clients, the client server, and also the community recognition server. The client server basically has the responsibility of getting the perception data and then sending them to the blockchain network to be uploaded, and also to the community detection server. Additionally, label data are also produced when the user links the dependency chain. The community detection server has the responsibility of gathering entire label data and carrying out the community detection technique, as well as

producing the community detection outcomes, and also sending such data to the blockchain network. When the community identification outcomes have been successfully saved to the blockchain database, the customer executes the smart contract in order to search for divided information from the community.

### 2.1.4. Blockchain Layer

Since the blockchain layer is made based on the hyperledger fabric structure, it consists of confirmation, order, and also committer nodes. The confirmation node has the responsibility of ensuring the transaction is offered through the customer server. The order node has the responsibility of classifying and packing the transactions into the blocks. The committer node also has the responsibility of validating and adding blocks into the database's blockchain. The common transaction procedure in the fabric is designed as the following stages:

- The customer side makes the transaction offer.
- The transaction execution is simulated by the node.
- The customer transmits the transaction into the consensus service.
- The customer orders the transactions with consensus, create new blocks, and renders the transactions.

In this article, the data-sharing structure applies to this transaction procedure in the phase of data sharing.

### 2.2. FDIA

The invaders attack the communication network, where invaders perfuse false data to messages or mensuration, which is shown in Figure 2. The FDI invader is able to perfuse the predetermined attack vector a by manipulating specific mensuration or messages. An invader is able to perfuse an attack vector to compromise the main mensuration which is shown in Equation (1); $e$ represents the error vector; $z$ defines the vector of measurements containing measurement readings from the sensor and ISO; $H$ defines the Jacobian matrix with respect to $x$.

$$\hat{z}_a = z + a = Hx + a + e \tag{1}$$

where: $a = (a_1, a_2, \ldots, a_m)^T \cap a \neq 0$. According to [14], the attack vector a is able to be adjusted by Equation (2).

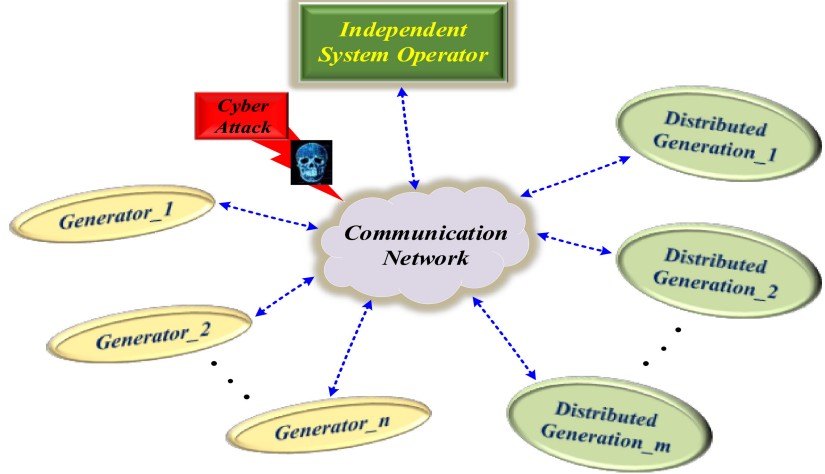

**Figure 2.** False data injection attack.

$$a = Hc, \tag{2}$$

where: $c = (c_1, c_2, \ldots, c_n)^T \cap c \neq 0$ is an arbitrary vector. The estimated state vector $\hat{x}$ is altered in Equation (3) with the false data injection.

$$\hat{x} = \left(H^T \Lambda H\right)^{-1} H^T \Lambda z_a = \hat{x} + c, \tag{3}$$

However, the residue $r$ stays unaltered after attack and it still less than threshold $\tau$;

$$r_a = ||\hat{z}_a - H\hat{x}_a||_2 = ||z + a - H(\hat{x} + c)||_2 = ||z - H\hat{x} + a - Hc)||_2 = ||z - H\hat{x}||_2 = r, \tag{4}$$

So the FDI invaders are able to circumvent and be undiscoverable to the traditional dab data detection.

## 3. Problem Theory

Limit pricing is one of the most well-known procedures of pricing congestion transmission networks, which is derived from the cost limit. The limit pricing is equal to the ratio of the increase in production costs to the increase in one megawatt of system load. If the limit pricing is calculated for increasing the load in a particular bus, it is called the node or local price, and if it is calculated for increasing the load in a particular area, it is called the regional price [41]. Node values for all busbars and regional prices for all zones will be equal when there is no network congestion.

Under normal power system conditions (lack of congestion and loss), logical marginal pricing (LMP) is the same in all busbars and there is a financial balance in the network busbars, in which case, this uniform price (called market clearing price (MCP)), as the purchase price and electricity sales, is used throughout the network, but when network congestion occurs, the LMP will be different on all busbars, which must be fixed using the methods mentioned in the corrective clogging management to fix the LMP busbars (fix the congestion of transmission lines).

A good way to manage network congestion is to put the market on a pricing basis. In this type of pricing, the ISO receives voluntary offers from market participants, selects the best solution with the lowest price, and finally performs the best load distribution; ultimately, all transmission line limitations are met and the system is balanced at the lowest price. Local price limits are obtained. In this paper, DC optimal load distribution (network losses are not considered) is used to manage clogging combined with social welfare (SW) maximization. To solve the DC optimal power flow (DCOPF) problem, equations have been used in the generalized algebra modeling system (GAMS) program. The discontinuous nonlinear program (DNLP) solver is used to solve the nonlinear program. The constraints of optimization include real equations of real power in all network buses, power transmission limitations, bus voltage limits, and production limits. The optimization of this issue is the amount of production of each generator (G) and the local limit price in all network buses. In this paper, in order to eliminate line congestion, with optimal use of distributed generation (DG), LMP stabilization of network buses in a certain value has been established. By stabilizing the local price limit of network buses in a fixed amount, the creation of market power by expensive generators is prevented. Under these conditions, the market is approaching full competition, and in general, the disadvantages caused by network congestion will be eliminated.

*Formulation*

The consumer interest function and the producer cost function are in accordance with relationships (5) to (8), respectively, in dollars per hour [41].

$$B_{dj}\left(P_{dj}\right) = -\frac{1}{2}c_{dj}P_{dj}{}^2 + d_{dj}P_{dj} + m_j \tag{5}$$

$$C_{gi}\left(p_{gi}\right) = \frac{1}{2}a_{gi}P_{gi}{}^2 + b_{gi}P_{gi} + m_i \tag{6}$$

$$C_k\left(P_{DG_k}\right) = \frac{1}{2}a_{DGk}P_{DGk}^2 + b_{DGk}P_{DGk}^2 + c_{DGk} \tag{7}$$

$$C_w = \gamma_w \times d_{wind} \times P_w \tag{8}$$

In these relations, $c_{dj}$ and $d_{dj}$ are slope and width from the origin of the uniform curve of $j - th$ consumer demand, $a_{gi}$ and $b_{gi}$ are slope and width from the origin of the uniform curve suggested by the generator, $m_j$ and $m_i$ are constant coefficients of profit and consumption functions of the $j - th$ generator and $i - th$ generator, $P_{dj}$ and $P_{gi}$ are the real power of the $j - th$ consumption and $i - th$ generator, $C_k(P_{DG_k})$ provides the cost function of the $k - th$ DG number, $N_m$ is the number of DGs connected to the network and $P_{DGk}$ represents the active power generation of the $k - th$ DG number, $C_w$ represents the cost of wind turbine production in each scenario, $d_{wind}$ is the recommended price of wind turbine per MW/h of power generation, $P_w$ gives the production power of the wind turbine unit in each scenario (MW), $w$ is considered as low and high scenarios for the power production of wind turbines in the set of $\Omega$, and $\gamma_w$ represents the probabilities for the two scenarios of wind power production, 0.4 and 0.6, that are chosen for the low and high scenarios, respectively.

The formulation of the problem of density management in reconstructed power systems is the maximization of social welfare by considering the power balance constraints and the density of transmission lines. Mathematically, the objective function of the problem (maximizing social welfare) is a nonlinear relation (9):

$$\text{Max } SW = \sum_{i=1}^{N_d} B_{di}(P_{di}) - \sum_{i=1}^{N_g} C_{gi}(P_{gi}) - \sum_{k=1}^{N_m} C_k(P_{DGk}) - \sum_{w \in \Omega} \gamma_w \times d_{wind} \times P_w \tag{9}$$

According to the formulation in relation to (9), the objective function, which is social welfare, is equal to the total profit of consumers minus the total cost of producers, which should be maximized according to the ISO view of this objective function. The equal and unequal constraints are given below. The profit of each manufacturer is obtained according to the following relationship:

$$\text{profit}_q = (LMP_{q\_n} \times P_{q\_n}) - C_q \tag{10}$$

where: $\text{profit}_q$ is the profit of the producer $q$, $LMP_{q\_n}$ is the local limit pricing in the $n - th$ bus where the $q - th$ producer is located, $P_{q\_n}$ is the production capacity of the q $-$ th producer in the $n - th$ bus, and $C_q$ is the cost of the q $-$ th producer. Hence, we have:

Power balance constraints as:

$$P_{gi} + P_{DGi} + P_W - P_{di} = \sum_{j=1}^{N} \frac{1}{x_{i-j}} (\delta_i - \delta_j), \text{ for } i \in u_{DG} \tag{11}$$

Maximum power limitation as:

$$\left| Pl_{i-j} \right| \leq Pl_{i-j}^{max, \, l} \tag{12}$$

Range of variables as:

$$0 \leq P_{gi} \leq P_{gi}^{max, \text{ for } g} \tag{13}$$

$$0 \leq P_{DGk} \leq P_{DG}^{max} \tag{14}$$

$$0 \leq P_{dj} \leq P_{dj}^{max, \text{ for } d} \tag{15}$$

$$\delta_i^{min_i \, i^{max} \text{ for}} \tag{16}$$

In these relations, $N$ and $N_L$ are the number of system busbars and the number of lines, respectively. $\delta_i$ is voltage angle in the $i - th$ busbar; $x_{i-j}$ gives the inductive reactance of the connecting line series amongst $i$ and $j$ buses; $u_{DG}$ provides the network DG set and $P_{DGk}^{max}$ is the operating rate of the $k - th$ DG. $Pl_{i-j}$ and $Pl_{i-j}^{max}$ are the active power and maximum active power at the connection line between the buses $i$ and $j$, respectively.

$P_{gi}{}^{max}$ and $P_{dj}{}^{max}$ represent maximum values of $P_{gi}$ and $P_{dj}$. $\delta_i^{min}$ and $\delta_i^{max}$ are the minimum and maximum values of $\delta_i$. The local limit value is also obtained from the power balance equilibrium in each bus.

Information of line parameters, generator cost coefficients, and consumption rates in different busbars is given in Tables 1–3, respectively.

**Table 1.** Grid line details.

| Line | From | To | X (p.u) | $Pl^{max}$ (MW) |
|------|------|-----|---------|------------------|
| 1 | 1 | 2 | 0/0592 | 60 |
| 2 | 1 | 5 | 0/2230 | 100 |
| 3 | 2 | 3 | 0/1980 | 100 |
| 4 | 2 | 4 | 0/1763 | 100 |
| 5 | 2 | 5 | 0/1739 | 50 |
| 6 | 3 | 4 | 0/1710 | 50 |
| 7 | 4 | 5 | 0/0421 | 100 |
| 8 | 4 | 7 | 0/2091 | 50 |
| 9 | 4 | 9 | 0/5562 | 50 |
| 10 | 5 | 6 | 0/2520 | 50 |
| 11 | 6 | 11 | 0/1989 | 50 |
| 12 | 6 | 12 | 0/2558 | 50 |
| 13 | 6 | 13 | 0/1303 | 50 |
| 14 | 7 | 8 | 0/1762 | 100 |
| 15 | 7 | 9 | 0/1100 | 100 |
| 16 | 9 | 10 | 0/0845 | 50 |
| 17 | 9 | 14 | 0/2704 | 50 |
| 18 | 10 | 11 | 0/1921 | 50 |
| 19 | 12 | 13 | 0/1999 | 50 |
| 20 | 13 | 14 | 0/3480 | 50 |

**Table 2.** Grid generator details.

| Gen No. | Bus No. | $P_g^{max}$(MW) | a (USD/(MWh)$^2$) | b (USD/MWh) | m (USD/h) |
|---------|---------|------------------|---------------------|--------------|------------|
| 1 | 1 | 250 | 0/43 | 20 | 0 |
| 2 | 2 | 200 | 0/25 | 20 | 0 |
| 3 | 3 | 60 | 0/01 | 40 | 0 |
| 4 | 6 | 50 | 0/01 | 40 | 0 |
| 5 | 8 | 60 | 0/01 | 40 | 0 |

**Table 3.** Consumption details in busbars.

| Load No. | Bus No. | $P_d$ (MW) |
|----------|---------|-------------|
| 1 | 2 | 8/17 |
| 2 | 3 | 42/89 |
| 3 | 4 | 56/05 |
| 4 | 5 | 26/52 |
| 5 | 6 | 35/6 |
| 6 | 9 | 26/26 |
| 7 | 10 | 12/65 |
| 8 | 11 | 39/68 |
| 9 | 12 | 11/8 |
| 10 | 13 | 10/7 |
| 11 | 14 | 33/91 |

In this work, it is assumed that four distributed generation units without uncertainty and one wind turbine with two high and low production scenarios, according to Table 4, are connected to the 14-bus IEEE test network. The specifications of the cost function of

these units and their capacity and installation location are given in Table 4. Additionally, the network of 14 buses that the IEEE studied is shown in Figure 3.

**Table 4.** Distributed generation (DG) unit details.

| DG | Bus No. | $P_{DG}^{max}$ (MW) | b_DG (USD/MWh) |
|---|---|---|---|
| DG1 | 14 | 20 | 30 |
| DG2 | 12 | 30 | 30 |
| DG3 | 9 | 20 | 30 |
| DG4 | 4 | 25 | 30 |
| wind | 10 | 5,20 | 15 |

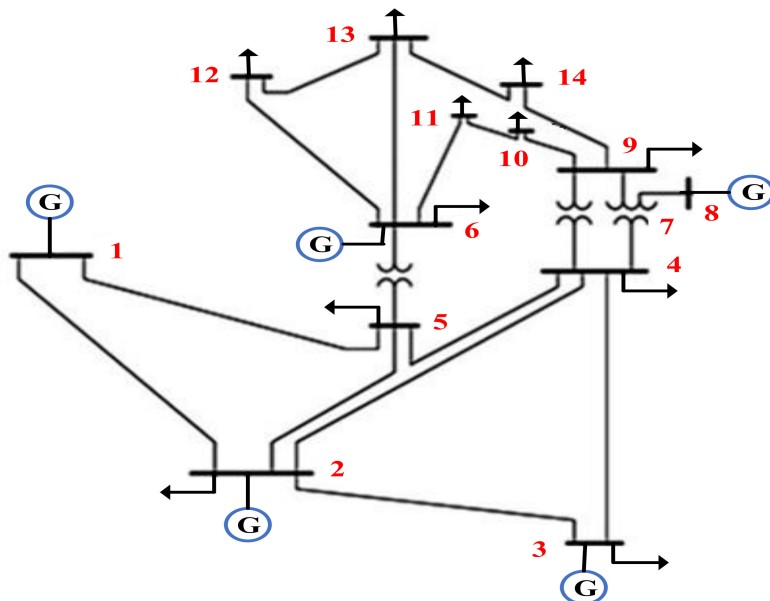

**Figure 3.** The studied network of 14 buses by the IEEE.

## 4. Simulation Results and Analysis

The studied network is the 14-bus IEEE network; the network consists of 12 loads, five generators, one uncertainty (wind) source, and three distributed generation (no uncertainty) sources for an hour of the day ahead of the electricity market. It is assumed that the wind would work in two scenarios: high and low. Wind generation in the low scenario is 5 MW and in the high scenario, it is 20 MW. The independent system operator (ISO), as an upstream entity, wants distributed generation sources (DGSs) available in the network to manage network congestion and maximize the social welfare and, ultimately, the ISO contract with the DGSs corresponds to their optimal generation for an hour of the day ahead. In the meantime, the wind turbine is presented as a balancing generator for the system. Without DGSs on the network, locational marginal prices (LMPs) are different in buses; in this case, there is congestion in the network. As shown in Figure 4, with the presence of DGSs, these prices are stabilized and network congestion is managed. As a result, social welfare is increased and operating costs are reduced.

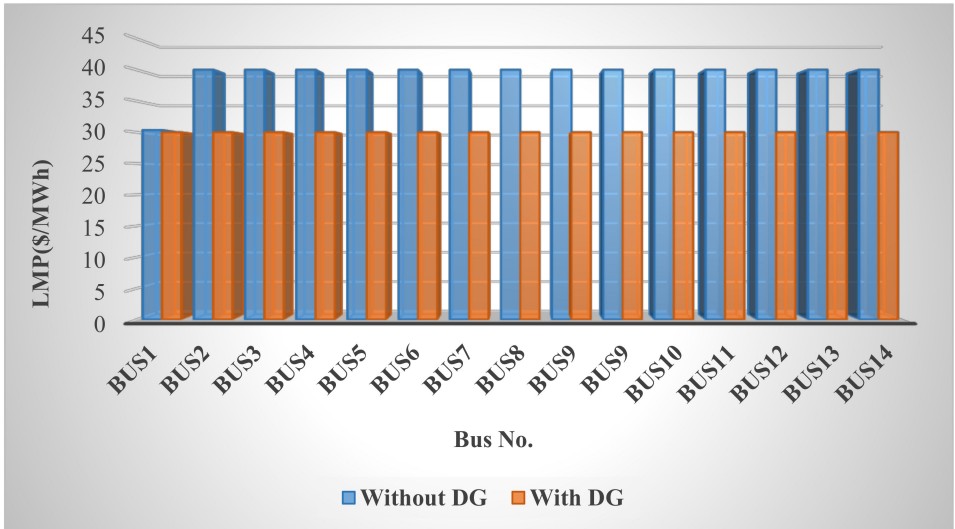

**Figure 4.** Locational marginal prices (LMPs) in different buses, with or without distributed generation sources in the network.

Next, it is been shown what will happen to the power system and electricity market situation with DGSs on the network by injecting different cyber-attacks (false data injection attacks), including load increase (LI), load decrease (LD), DGS price changes (DGPCH), and generator price changes (GPCH). In the FDI attacks, the attacker is able to access the data of the communication links, sensors, local controllers, and central control units so, to simulate the FDI attack, it has been assumed that the attacker can manipulate the data, therefore at the time of the attack, the data has been manipulated to show the attack outcomes.

*4.1. Scenario 1: Normal*

In this case, it is assumed that the network is in operation with the presence of DGSs, and no false data injection attack has been performed on the network load and unit prices. According to Table 5, the optimal production and profit of each of the operating units in the 14-bus network is given. In this case, cheap generators 1 and 2 in both cases are wind production in the production line, while the more expensive generators 3, 4 and 5 are not used. Figure 5 shows LMPs on different buses without network attacks.

**Table 5.** Status of units in normal mode.

| Unit | Bus No. | Generation (MW) | | Profit (USD) | |
|---|---|---|---|---|---|
| | | Low | High | Low | High |
| G1 | 1 | 115.314 | 100 | 2878.451 | 2439.277 |
| G2 | 2 | 20 | 20 | 500 | 500 |
| G3 | 3 | 0 | 0 | 0 | 0 |
| G4 | 6 | 0 | 0 | 0 | 0 |
| G5 | 8 | 0 | 0 | 0 | 0 |
| | Total | | | 3378.451 | 2939.277 |
| Unit | Bus No. | Generation (MW) | | Profit (USD) | |
| | | Low | High | Low | High |
| DG1 | 14 | 15.604 | 6.054 | 468.123 | 468.123 |
| DG2 | 12 | 0 | 4.613 | 0 | 0 |
| DG3 | 9 | 6.470 | 2.535 | 194.103 | 194.103 |
| DG4 | 4 | 4.108 | 4.108 | 123.231 | 123.231 |
| | Total | | | 785.457 | 519.309 |

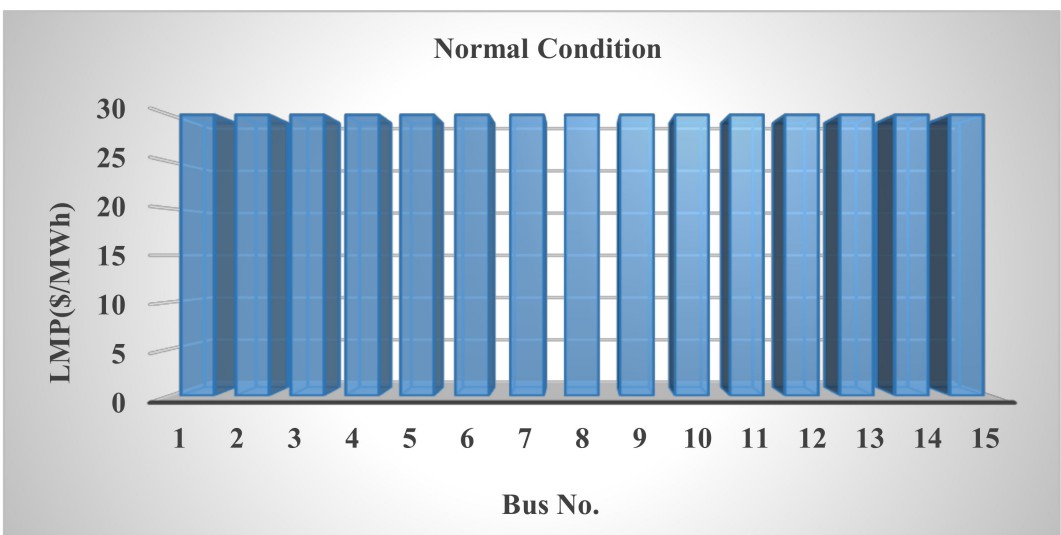

**Figure 5.** LMPs on different buses without network attacks.

DGSs are also in operation, as shown in Table 5. It should be noted that these optimal results will lead to optimal network status. Bus node prices are reduced and stabilized, network congestion is managed, and network social welfare is maximized. As a result, ISO can contract with distributed generation units in accordance with these optimal production values. The results are obtained by the powerful GAMS optimization software. In this case, social welfare is 3554.

*4.2. Scenario 2: Incremental Attack on Load*

Sometimes the attacker tries to break the market and disrupt the power supply for various reasons. One of these types of attacks is the false data injection attack on the smart power network components, such as loads, measurements, detectors, and sensors.

In Scenario 2, it is assumed that the attacker has access to the loads of one hour from the day ahead of the market and by virtually changing the loads by 1.8 times of the main load, the attacker can change the production conditions and profit of the units in the market and create congestion in the network. In this case, LMPs increase in all buses, so this price in bus 1 is different from other buses, in which case it can be said that there is congestion in the network. According to Figure 6, as the load increases, according to Table 5, all units increase their production. DGSs generate electricity at their maximum capacity, and cheap and even expensive generators are on the production line. Given that in reality there has been no increase in load and the attacker has changed the network load information, additional production takes place in the network, which will lead to serious damage to the network and also increase production costs and reduce social welfare. In this case, social welfare is 3148.

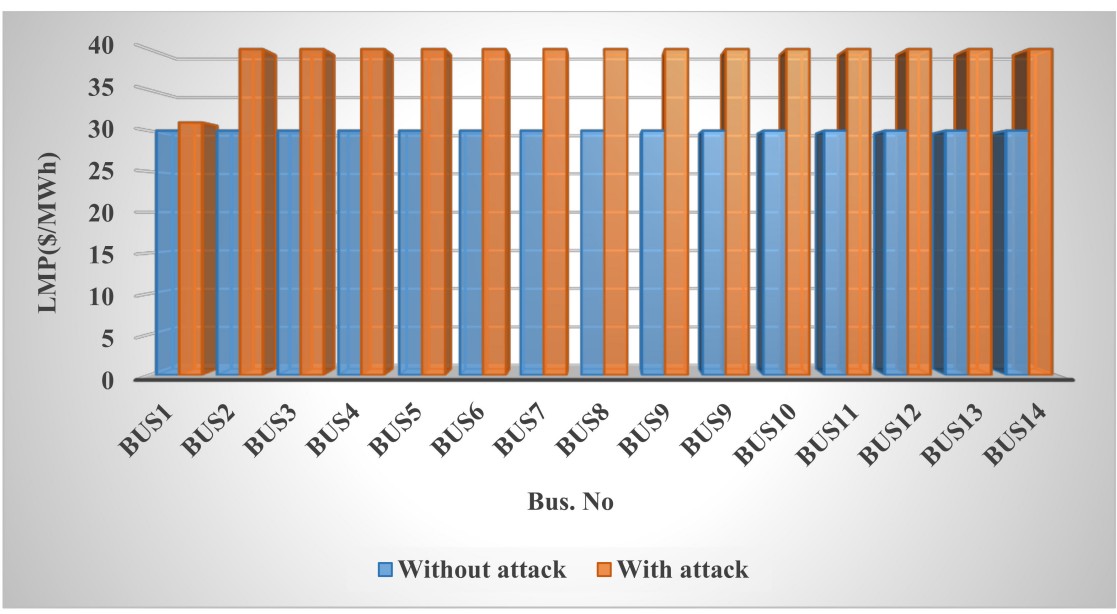

**Figure 6.** LMPs at different buses in the case of an incremental load attack.

### 4.3. Scenario 3: Decreasing Attack on Load

In scenario 3, it is assumed that the attacker has access to the loads of one hour from the day ahead of the market and by virtually reducing the loads by 0.5 times of the main load, the attacker can change the production conditions and profit of the units in the market.

The LMPs have dropped, as shown in Figure 7, but because this is not the case in reality and the load has not been reduced, generators have minimized their production and DGSs have very little production (according to Table 6), which causes failure to provide real loads and a large reduction in unit profits, especially production units, will be dispersed. Additionally, due to the imbalance of load and production in the network, blackouts occur in various sections of the grid. Social welfare in this case is 2536, which is a decrease compared to the normal state.

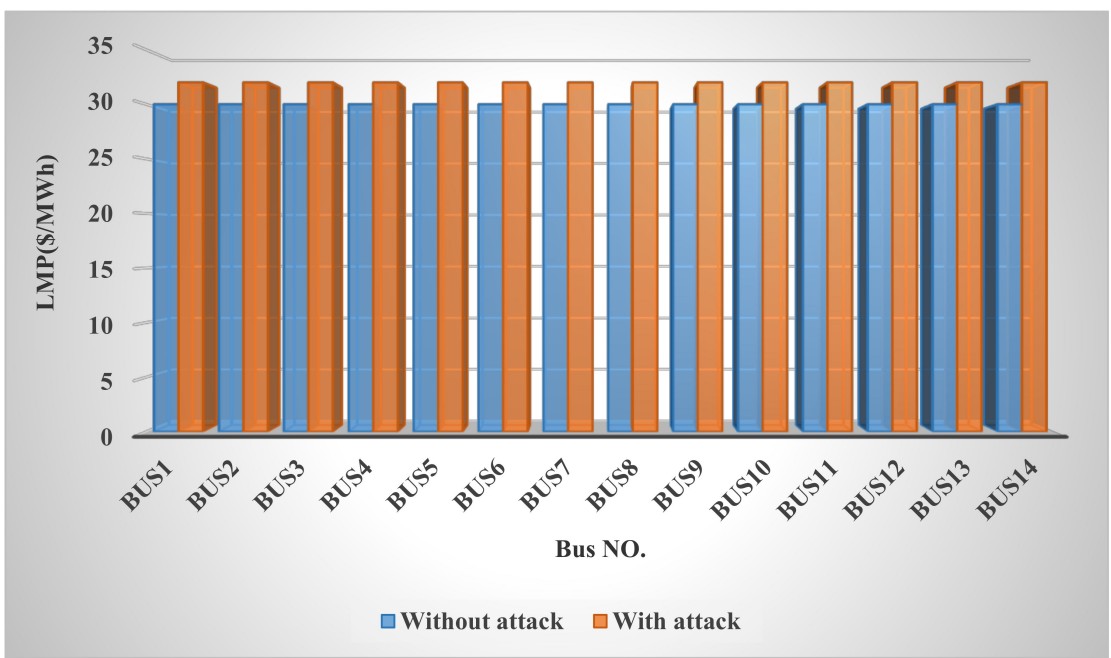

**Figure 7.** LMPs at different buses in the case of a reduced load attack.

**Table 6.** Status of units in incremental load attack mode.

| Unit | Bus No. | Generation (MW) | | Profit (USD) | |
|------|---------|------|------|------|------|
| | | Low | High | Low | High |
| G1 | 1 | 115.314 | 100.314 | 2878.451 | 2439.277 |
| G2 | 2 | 40.479 | 40.479 | 1219.212 | 1219.212 |
| G3 | 3 | 11.972 | 11.972 | 480.303 | 480.303 |
| G4 | 6 | 11.972 | 11.972 | 387.669 | 387.669 |
| G5 | 8 | 11.972 | 11.972 | 387.669 | 387.669 |
| Total | | | | 5538.572 | 5099.398 |
| Unit | Bus No. | Generation (MW) | | Profit (USD) | |
| | | Low | High | Low | High |
| DG1 | 14 | 20 | 20 | 600 | 600 |
| DG2 | 12 | 30 | 30 | 900 | 900 |
| DG3 | 9 | 20 | 20 | 600 | 600 |
| DG4 | 4 | 25 | 25 | 750 | 750 |
| Total | | | | 2850 | 2850 |

*4.4. Scenario 4: Attacking Generator Bid Prices*

It is supposed that the attacker has access to the generator's suggested prices. The attacker can change the suggested prices of the generators (in this scenario, the prices of cheap and expensive production units were shifted together). In this case, according to Table 7, cheap generators and even DGSs have no production, and this will seriously damage the interests of these units. In this case, the power supply of the network will only be the responsibility of expensive generators and will disrupt the financial balance of the market. Table 8 provides the status of units in attack mode at generator prices.

**Table 7.** Status of units in the case load reduction attack mode.

| Unit | Bus No. | Generation (MW) | | Profit (USD) | |
|------|---------|------|------|------|------|
| | | Low | High | Low | High |
| G1 | 1 | 58.467 | 58.467 | 1316.427 | 1434.832 |
| G2 | 2 | 10.063 | 10.063 | 226.580 | 246.959 |
| G3 | 3 | 0 | 0 | 0 | 0 |
| G4 | 6 | 0 | 0 | 0 | 0 |
| G5 | 8 | 0 | 0 | 0 | 0 |
| Total | | | | 1543.007 | 1681.791 |
| Unit | Bus No. | Generation (MW) | | Profit (USD) | |
| | | Low | High | Low | High |
| DG1 | 14 | 0.002 | 0 | 0.006 | 0 |
| DG2 | 12 | 0.002 | 0 | 0.006 | 0 |
| DG3 | 9 | 0.002 | 0 | 0.006 | 0 |
| DG4 | 4 | 0.002 | 0.002 | 0.006 | 0.006 |
| Total | | | | 0.024 | 0.006 |

**Table 8.** Status of units in attack mode at generator prices.

| Unit | Bus No. | Generation (MW) | |
|------|---------|------|------|
| | | **Low** | **High** |
| G1 | 1 | 0 | 0 |
| G2 | 2 | 0 | 0 |
| G3 | 3 | 49.853 | 49.853 |
| G4 | 6 | 49.853 | 49.853 |
| G5 | 8 | 49.853 | 49.853 |
| **Unit** | **Bus No.** | **Generation (MW)** | |
| | | **Low** | **High** |
| DG1 | 14 | 0.07 | 0 |
| DG2 | 12 | 0 | 0 |
| DG3 | 9 | 0 | 0 |
| DG4 | 4 | 0.003 | 0 |

*4.5. Scenario 5: Attack on Distributed Generation Prices*

In this case, also, the attacker manipulates the proposed prices of DGSs (price increases by 1.4 times the original prices). As a result, according to Table 9, the generation of DGSs reaches zero and does not benefit them. Thus, operating costs increase and social welfare decreases (3320).

**Table 9.** Status of units in attack mode at DG prices.

| Unit | Bus No. | Generation (MW) | |
|------|---------|------|------|
| | | **Low** | **High** |
| G1 | 1 | 115.314 | 100.314 |
| G2 | 2 | 40.023 | 40.023 |
| G3 | 3 | 0.574 | 0.574 |
| G4 | 6 | 0.574 | 0.574 |
| G5 | 8 | 0.574 | 0.574 |
| **Unit** | **Bus No.** | **Generation (MW)** | |
| | | **Low** | **High** |
| DG1 | 14 | 0 | 0 |
| DG2 | 12 | 0 | 0 |
| DG3 | 9 | 0 | 0 |
| DG4 | 4 | 0 | 0 |

*4.6. Generation of Units in Different Scenarios*

Figure 8 shows the production of generators in different scenarios. This figure shows that all generators in scenario 2 due to increased load are on the network production line. The lowest generator output was related to the load reduction data attack scenario. Figure 9 also shows the production of DGSs in different scenarios. In this case, the highest production is related to scenario 1.

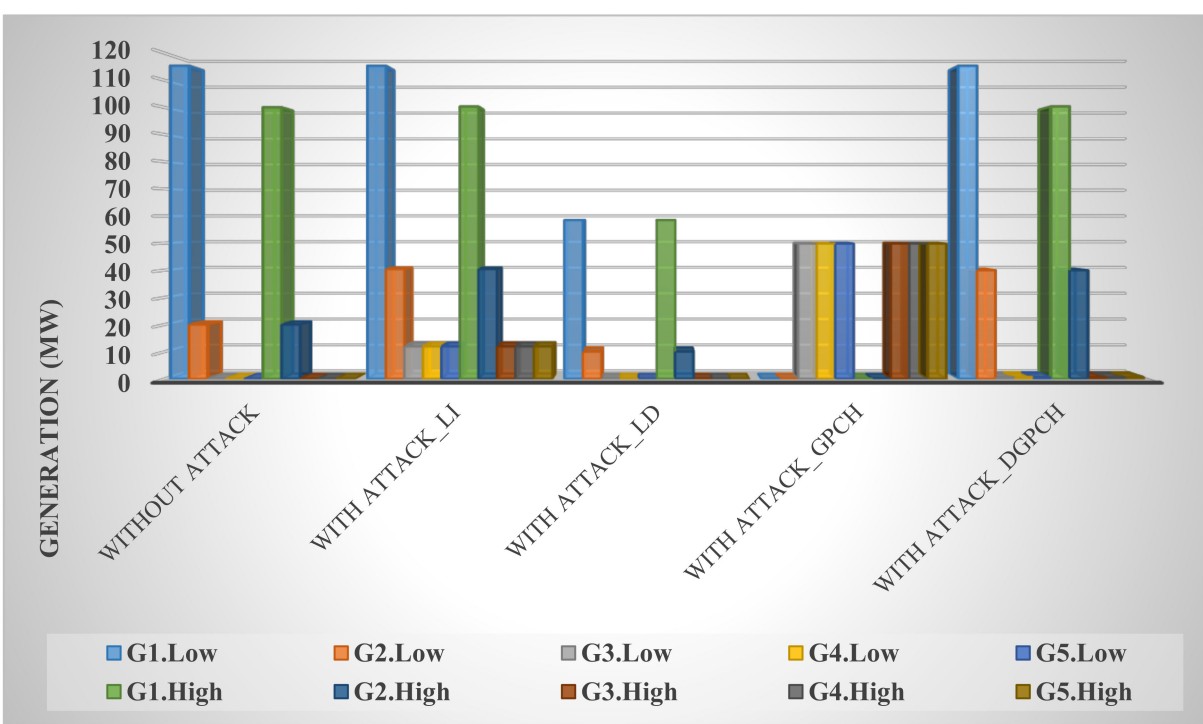

**Figure 8.** Production of generators in different scenarios.

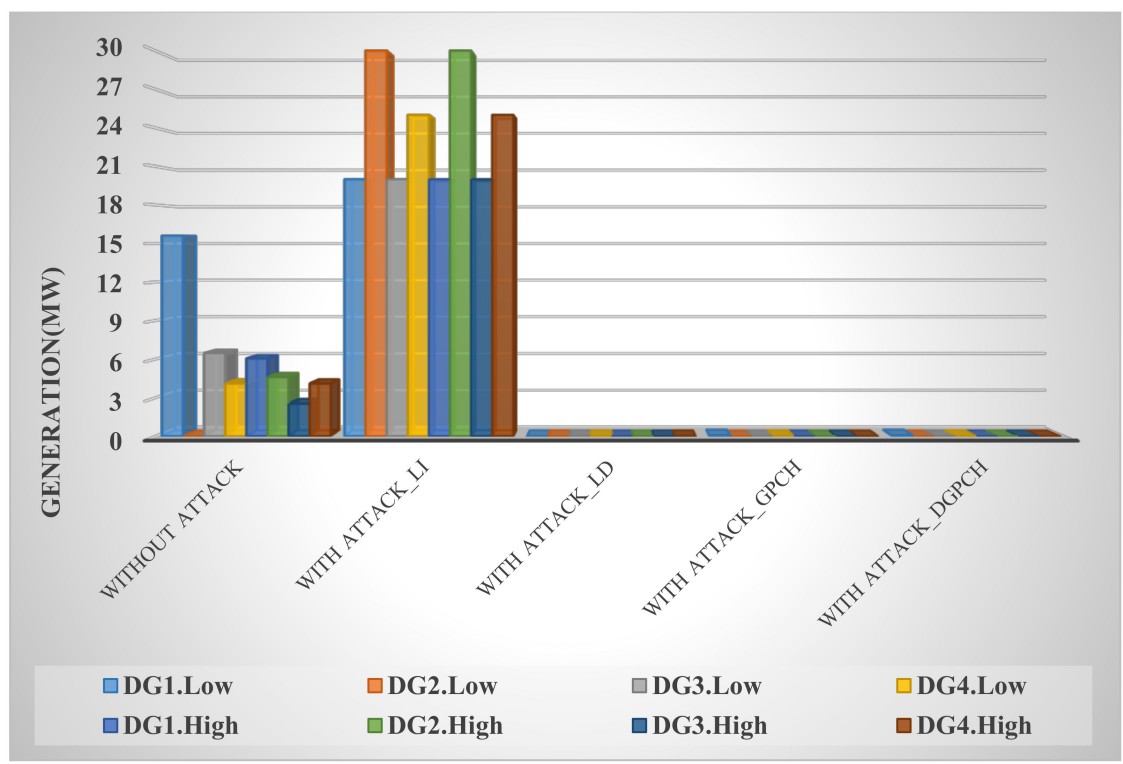

**Figure 9.** Production of distributed generation sources (DGSs) in different scenarios.

*4.7. Implementation of Blockchain for Secure Exchange of Messages amongst ISO and Under-Operating Agents*

In this scenario, in order to secure the data exchanged between ISO and under-operating agents, the blockchain technique has been used to exchange information, and for

this purpose, unit prices are sent to ISO along with the production rate based on the system conditions. Confirmation of the validity of the sent data in the blockchain platform and non-destructive manipulation of information, such as FDI attack, and the amount of production of units in the desired time frame are sent to the units in the blockchain platform and then the units, after validating the sent data by ISO, produce the desired amount of power, which makes the data safe to send and any attack based on the messages is detected. It should be noted that the main objectives of the independent system operator, as in the above institution, should be met, such as reducing the LMP of the buses (reducing the congestion of transmission lines) and maximizing the social welfare of the network. DGSs should be properly closed. Table 10 shows examples of exchanged messages between ISO and under-operating agents based on blockchain technology.

**Table 10.** Transaction blockchain description.

| Block Index | | Block Information | | |
|---|---|---|---|---|
| Index | | 1 | | **Description** |
| Time | | 1 | | |
| Transaction Message | Sender | Receiver | Power (MW) | |
| (Data) | ISO | DG3 | 6.470 | |
| Previous HA | | 6ace5c77c9f1abba4f25f81f5557ca3d | | |
| Self HA | | 7ab52efec4674cbf34f97e2a72d5e753 | | |
| Index | | 2 | | **Description** |
| Time | | 2 | | |
| Transaction Message | Sender | Receiver | Power (MW) | |
| (Data) | ISO | DG4 | 4.108 | |
| Previous HA | | 7ab52efec4674cbf34f97e2a72d5e753 | | |
| Self HA | | 266477dd8561492cfa2bef961485b8a4 | | |
| Index | | 4 | | **Description** |
| Time | | 4 | | |
| Transaction Message | Sender | Receiver | Power (MW) | |
| (Data) | ISO | G1 | 115.314 | |
| Previous HA | | 098e81b5609be3f39bdb61c2e6d2c67d | | |
| Self HA | | de3a8a139ecee8aaf023fc914417d748 | | |
| Index | | 5 | | **Description** |
| Time | | 4 | | |
| Transaction Message | Sender | Receiver | Power (MW) | |
| (Data) | ISO | G1 | 80 | Cyber-attack has |
| Previous HA | | 098e81b5609be3f39bdb61c2e6d2c67d | | occurred |
| Self HA | | 029e6f1af9dacfe2f3e01fda58634a00 | | |
| Index | | 6 | | **Description** |
| Time | | 5 | | |
| Transaction Message | Sender | Receiver | b (USD/MWh) | |
| (Data) | G4 | ISO | 40 | |
| Previous HA | | de3a8a139ecee8aaf023fc914417d748 | | |
| Self HA | | 3490acda48423d504c295da91fa2aa92 | | |
| Index | | 7 | | **Description** |
| Time | | 5 | | |
| Transaction Message | Sender | Receiver | b (USD/MWh) | |
| (Data) | G4 | ISO | 20 | Cyber-attack has |
| Previous HA | | de3a8a139ecee8aaf023fc914417d748 | | occurred |
| Self HA | | 706ef584b4a6529f9af410c950fcecc9 | | |

The generic blockchain relevant to ISO is demonstrated in Table 10 and shows that the information is in accordance with the specific blockchain of ISO. This procedure is able to aid the recovery of the information contained in a cyber-attack or package in private and public blockchains.

Nevertheless, it is noteworthy that the output powers of agent generation and other data, as well as the costs of the generation of every DG and G, do not exist in the blockchain, which is able to raise the privacy and the security of the messages and network.

The transaction blocks are presented by Table 10. Pursuant to this table, for example, at t = 1, DG3 gets a message from ISO. In addition, the value of the provided power in megawatts, as well as the generating DG, are demonstrated in this table. Like the specific blockchain, any block includes the blockchain hash algorithm (HA), recognized as the self HA, that is able to chain to the prior block through utilizing the prior HA. Plus, if a cyber-attack has happened in the message (see indexes 7 and 5), the HA is altered and the HA is not the similar, so the multiple message is defined.

## 5. Conclusions

DG resources have been applied to reduce LMPs and maximize the social welfare of the network to reach ISO's main objectives. It has been shown that by applying DGs in power grids and stabilizing the LMP of the busbars, the congestion was managed and the production of more expensive units was minimized. Additionally, the social welfare of the network was maximized for the seasons under contract with DG. The blockchain technology has been used to secure messages and exchanged data between ISO and under-operating agents.

The results demonstrated that ISO can modify its polices and use the potential and benefits of DGSs to increase social welfare and reduce line density by concluding contracts in accordance with the production values given. In addition to the cyber security reinforcement, the considered policy sample is decentralized, transparent, and secured, and is able to decrease the risks to the network, remove the financial spoof, and reduce the whole cost of operation.

The outcomes of simulation on a trial system confirmed the great effectiveness and performance of the considered policy frame, particularly in the presence of a cyber-attack where the information is not available for outside unwarranted parts of the system. This is chiefly because the HAs are altered in any repeat.

**Author Contributions:** M.D. and M.G. proposed the idea, developed the model, and performed the simulation works and also wrote the paper. T.N. and A.K.-F. led the project. F.T.-H., M.S. and N.G. were in charge of reviewing and editing the paper. This work was conducted under the supervision of T.N., F.T.-H. All authors have read and agreed to the published version of the manuscript.

**Funding:** This research received no external funding.

**Data Availability Statement:** Data is contained within the article.

**Conflicts of Interest:** The authors declare no conflict of interest.

## Nomenclature

| | |
|---|---|
| **z** | Vector of measurements |
| **H** | Jacobian matrix |
| **a** | Attack vector |
| **c** | Arbitrary vector |
| **x̂** | Estimated state vector |
| **r** | Residue |
| **τ** | Threshold |
| $\mathbf{c_{dj}}$ | Slope from the origin of the uniform curve of $j-$th consumer demand |
| $\mathbf{d_{dj}}$ | Width from the origin of the uniform curve of $j-$th consumer demand |
| $\mathbf{a_{gi}}$ | Slope from the origin of the uniform curve suggested by the generator |
| $\mathbf{b_{gi}}$ | Width from the origin of the uniform curve suggested by the generator |

| | |
|---|---|
| $m_j$ | Constant coefficients of profit and consumption functions of $j-$ th G |
| $m_i$ | Constant coefficients of profit and consumption functions of $i-$ th G |
| $P_{dj}$ | Real power of $j-$ th consumption |
| $P_{gi}$ | Real power of $i-$ th generator |
| $C_k(P_{DG_k})$ | Cost function of $k-$ th DG number |
| $N_m$ | Number of DGs connected to the network |
| $P_{DGk}$ | Active power generation of $k-$ th DG number |
| $C_w$ | Cost of wind turbine production |
| $d_{wind}$ | Recommended price of wind turbine |
| $P_w$ | Production power of wind turbine unit |
| $w$ | Considered as low and high scenarios for power production of wind turbine |
| $\gamma_w$ | Probabilities for the two scenarios of wind power production |
| $Profit_q$ | Profit of the producer q |
| $LMP_{q\_n}$ | Local limit pricing in $n-$ th bus where the $q-$ th producer is located |
| $P_{q\_n}$ | Production capacity of the $q-$ th producer in $n-$ th bus |
| $C_q$ | Cost of $q-$ th producer |
| $N$ | Number of system busbars |
| $N_L$ | Number of lines |
| $\delta_i$ | Voltage angle in $i-$ th busbar |
| $x_{i-j}$ | Inductive reactance of the connecting line series amongst i and j buses |
| $u_{DG}$ | Network DG set |
| $P_{DGk}^{max}$ | Operating rate of $k-$ th DG |
| $Pl_{i-j}$ | Active power at the connection line between the buses i and j |
| $Pl_{i-j}^{max}$ | Maximum active power at the connection line between buses i and j |
| $P_{gi}^{max}$ | Maximum values of $P_{gi}$ |
| $P_{dj}^{max}$ | Maximum values of $P_{dj}$ |
| $\delta_i^{min}$ | Minimum values of $\delta_i$ |
| $\delta_i^{max}$ | Maximum values of $\delta_i$ |
| **List of abbreviations** | |
| **DG** | Distributed generation |
| **FDIA** | False data injection attack |
| **ISO** | Independent system operation |
| **SGs** | Smart grids |
| **CPPS** | Cyber-physical power system |
| **DOS** | Denial of service |
| **FDI** | False data injection |
| **MG** | Micro-grid |
| **MGs** | Micro-grids |
| **DSO** | Distributed system operator |
| **IoT** | Internet of Things |
| **P2P** | Peer-to-peer |
| **PHEVs** | Plug-in hybrid electric vehicles |
| **GPS** | Global positioning system |
| **LMP** | Logical marginal pricing |
| **MCP** | Market clearing price |
| **SW** | Social welfare |
| **DCOPF** | DC optimal power flow |
| **GAMS** | Generalized algebra modeling system |
| **G** | Generator |
| **DNLP** | Discontinuous nonlinear program |
| **DGSs** | Distributed generation sources |
| **LI** | Load increase |
| **LD** | Load decrease |
| **DGPCH** | Distributed generation source price changes |
| **GPCH** | Generator price changes |
| **LMPs** | Locational marginal prices |
| **DGs** | Distributed generations |
| **HA** | Hash algorithm |

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
