# Peer review of "Blockchain-Based Securing of Data Exchange in a Power Transmission System Considering Congestion Management and Social Welfare"

_sustainability, doi:10.3390/su13010090_

Round 1

Reviewer 1 Report

In this paper authors proposed an application of blockchain in Power Transmission System.

The paper is well written and easy to follow however specific details of blockchain required for the experiment is missing and various non-related, general and extra informations are presented for blockchain system.

It is not clear how reader will believe that results presented through graphs are verifiable and correct. No implementation details are given. 

The blockchain concepts are taken from other papers, however authors tried to remove plagiarism but it's better to write sentences in your own language rather than copying and removing plagiarism.   

Author Response

Reviewer # 1:

In this paper authors proposed an application of blockchain in Power Transmission System.

Comment 1: The paper is well written and easy to follow however specific details of blockchain required for the experiment is missing and various non-related, general and extra information are presented for blockchain system.

Authors’ Response: Firstly, I would like to thank for your accurate attention and for your valuable recommendations. We are glad to hear the feedback and would like to thank you for giving us constructive feedback and suggestions to improve the paper. Thank you very much for your comment. In the revised version of the manuscript, the blockchain part is completely revised. Please see the paper.

Comment 2: It is not clear how reader will believe that results presented through graphs are verifiable and correct. No implementation details are given. 

Authors’ Response: Thank you very much for your valuable comment. The system details, problem formulation the objective function of the problem (maximizing social welfare) and constrains presented in 3th section “Formulation”. We stated in the paper that: to solve the DC Optimal Power Flow (DCOPF) problem, equations have been used in the generalized algebra modeling system (GAMS) program.

We added the follow part to the paper: In the FDI attacks, the attacker is able to access to the data of the communication links, sensors, local controllers and central control units so, to simulate the FDI attack, it has been assumed that the attacker can manipulated the data, therefore at the attacks time, the data has been manipulated to show the attack outcomes.

Comment 3: The blockchain concepts are taken from other papers, however authors tried to remove plagiarism but it's better to write sentences in your own language rather than copying and removing plagiarism.

Authors’ Response: Thank you very much for your comment. The blockchain concept is completely revised and write it in own language. Please see the paper.

At the end, we would like to express our special appreciation for spending your valuable time to review the manuscript.

Reviewer 2 Report

The subject matter of this article is current and important from both theoretical and practical points of view. The authors have presented the issue of congestion management in power transmission networks using the Blockchain technology under the threat of cyber attacks. They proposed a theoretical model of the system, which was verified on a test 14-bus network of the IEEE type with simulation methods. The results of these studies are interesting and have been presented convincingly, although I have a few criticisms, the inclusion of which would improve the readability of the article.

  • The lack of a list of abbreviations used in the work (e.g. MG in line 80) and symbols applied in equations 1 to 16 significantly hinders proper understanding of the text.
  • The graphs shown in Figures 4, 5 and 6 are presented in a uniform way, while in Figure 7 a different scale of the y-axis is adopted, which may mislead the reader when comparing all these graphs.
  • Section 5 - Conclusion is more of a summary of the article than a systematic ordering of the research results. I suggest changing the structure of this section in such a way that the reader can read what, according to the authors, is the main research achievement and what practical conclusions result from this research.
  • I think that the title of the work is not fully adequate to the content of the article, because I have not found any elements that would justify exposing the phrase "Social Welfare Enhancement" in the first place of the article title.

Author Response

Reviewer # 2:

The subject matter of this article is current and important from both theoretical and practical points of view. The authors have presented the issue of congestion management in power transmission networks using the Blockchain technology under the threat of cyber-attacks. They proposed a theoretical model of the system, which was verified on a test 14-bus network of the IEEE type with simulation methods. The results of these studies are interesting and have been presented convincingly, although I have a few criticisms, the inclusion of which would improve the readability of the article.

Comment 1: The lack of a list of abbreviations used in the work (e.g. MG in line 80) and symbols applied in equations 1 to 16 significantly hinders proper understanding of the text.

Authors’ Response: Firstly, I would like to thank for your accurate attention and for your valuable recommendations. We are glad to hear the feedback and would like to thank you for giving us constructive feedback and suggestions to improve the paper. Thank you very much for your valuable comment. The nomenclature and list of abbreviation table is added to the revised paper. Please see the paper.

Nomenclatures

Maximum values of

Vector of measurements

Maximum values of

Jacobian matrix

Minimum values of

Attack vector

Maximum values of

Arbitrary vector

List of abbreviation

Estimated state vector

Residue

DG

Distributed generation

threshold

FDIA

False data injection attack

Slope from the origin of the uniform curve of  consumer demand

ISO

Independent system operation

Width from the origin of the uniform curve of  consumer demand

SGs

Smart-grids

Slope from the origin of the uniform curve suggested by the generator

CPPS

Cyber-physical power system

Width from the origin of the uniform curve suggested by the generator

DOS

Denial-of service

Constant coefficients of profit & consumption functions of  G

FDI

False data injection

Constant coefficients of profit & consumption functions of  G

MG

Micro-grid

Real power of  consumption

MGs

Micro-grids

Real power of  generator

DSO

Distributed system operator

Cost function of  DG number

IoT

Internet of things

Number of DGs connected to the network

P2P

Peer-to-Peer

Active power generation of  DG number

PHEVs

Plug-in hybrid electric vehicles

Cost of wind turbine production

GPS

Global positioning system

Recommended price of wind turbine

LMP

Logical marginal pricing

Production power of wind turbine unit

MCP

Market clearing price

Considered as low and high scenarios for power production of wind turbine

SW

Social welfare

Probabilities for the two scenarios of wind power production

DCOPF

DC optimal power flow

Profit of the producer

GAMS

Generalized algebra modeling system

Local limit pricing in  bus where the producer of  is located

G

Generator

Production capacity of the producer  in  bus

DNLP

Discontinuous nonlinear program

Cost of  producing

DGSs

Distributed generation sources

N

Number of system bus-bars

LI

Load increase

Number of lines

LD

Load decrease

Voltage angle in  busbar

DGPCH

Distributed generation sources price changes

Inductive reactance of the connecting line series amongst  and  buses

GPCH

Generators price changes

Network DG set

LMPs

Locational marginal prices

Operating rate of   DG

DGs

Distributed generations

Active power at the connection line between the buses  and

HA

Hash algorithm

Maximum active power at the connection line between buses  and

Comment 2: The graphs shown in Figures 4, 5 and 6 are presented in a uniform way, while in Figure 7 a different scale of the y-axis is adopted, which may mislead the reader when comparing all these graphs.

Authors’ Response: Thank you very much for your comment. According to your suggestion, in the revised version of the manuscript, Figure 7 is edited. Please see the paper.

Figure 7. LMPs at different buses in the case of a reduced load attack

Comment 3: Section 5 - Conclusion is more of a summary of the article than a systematic ordering of the research results. I suggest changing the structure of this section in such a way that the reader can read what, according to the authors, is the main research achievement and what practical conclusions result from this research.

Authors’ Response: Thank you very much for your comment. The Conclusion is edited in the revised version of the paper. Please see the paper.

Comment 4: I think that the title of the work is not fully adequate to the content of the article, because I have not found any elements that would justify exposing the phrase "Social Welfare Enhancement" in the first place of the article title.

Authors’ Response: According to your suggestion, the title of the paper is edited in the revised paper and changed to ‘Securing Data Exchange in Power Transmission System Based on Blockchain Technology Considering Congestion Management and Social Welfare’.

At the end, we would like to express our special appreciation for spending your valuable time on our manuscript.
